computer modelling and simulation

opinion dynamics, network science, social networks

**Author for correspondence:**
Giacomo Livan
e-mail: g.livan@ucl.ac.uk

# The impact of noise and topology on opinion dynamics in social networks

Samuel Stern[1] and Giacomo Livan[1,2]

[1]Department of Computer Science, University College London, Gower Street, London WC1E 6EA, UK
[2]Systemic Risk Centre, London School of Economics and Political Sciences, Houghton Street, London WC2A 2AE, UK

 GL, 0000-0001-5412-6555

We investigate the impact of noise and topology on opinion diversity in social networks. We do so by extending well-established models of opinion dynamics to a stochastic setting where agents are subject both to assimilative forces by their local social interactions, as well as to idiosyncratic factors preventing their population from reaching consensus. We model the latter to account for both scenarios where noise is entirely exogenous to peer influence and cases where it is instead endogenous, arising from the agents' desire to maintain some uniqueness in their opinions. We derive a general analytical expression for opinion diversity, which holds for any network and depends on the network's topology through its spectral properties alone. Using this expression, we find that opinion diversity decreases as communities and clusters are broken down. We test our predictions against data describing empirical influence networks between major news outlets and find that incorporating our measure in linear models for the sentiment expressed by such sources on a variety of topics yields a notable improvement in terms of explanatory power.

## 1. Introduction

There is considerable value in understanding how opinions are formed, distributed and spread within a society, and how they evolve over time. Many of the decisions we make and actions we undertake are influenced by our opinions and beliefs. These cannot always be seen as deterministic, however, as they are continuously subject to change from a multitude of factors. Along with self-reflection and external information, social interactions are a significant contributor to how we form opinions. Businesses, for instance, need to know how and when their consumers' preferences evolve so they can adapt and better satisfy their

customers' needs. Governments and public servants need to be alert to their constituents' shifting values and adapt their policies to reflect these values.

Opinion dynamics is the study of how individuals form opinions as a result of social interactions [1]. The field has seen increased research interest over the past few decades, ostensibly due to the rise of online social media platforms and e-commerce. There are numerous problems of interest including whether, or how quickly, people's opinions converge to agreement [2,3], how to identify the most influential actors [4], and the effects that stubborn and zealous actors have on the system dynamics [5–7]. One of the important areas that is frequently overlooked is that of 'opinion diversity'. Opinion diversity captures how varied the set of opinions are within a society.

Modelling society as a social network provides a useful mathematical framework for understanding how opinion dynamics leads to differing outcome behaviours. However, in an environment where opinions may be stochastic, it is still unclear precisely what role network topology plays in the distribution of opinion. To gain a better understanding of this, we investigate the effect that varying network structure has on the diversity of opinions.

Depending on the context and how it is defined, discord and diversity among opinions within a collection of individuals can be viewed as either a positive or negative property. Greater issue diversity in politics (i.e. the extent to which the public is concerned with a wide range of issues), for example, mitigates the negative effects that party polarization has on the public's satisfaction with democracy [8]. In the media, diversity of news coverage provides the public with access to a broader range of issues, encouraging better-informed decisions about their community and quality of life [9,10]. However, diversity can also highlight conflict. In social media, for instance, we frequently see polarization, especially on politically motivated issues [11,12]. The types of scenarios where diversity is undesirable is in situations where there is a ground truth, such as whether climate change is a hoax or whether vaccines increase the likelihood of developing autism.

There is a growing body of literature looking at conflict within social networks, the bulk of which treats the topic as an optimization problem. Mackin & Patterson [13], for instance, propose a method for optimal leader placement, while Musco *et al.* [14], suggest an algorithm that optimizes the weights in a network as a means to minimize polarization. The results of such optimization methods can be valuable to network administrators, especially for 'social planning' purposes, such as making link recommendations. However, they are of limited practical utility when, as is often the case, administrators have neither the means nor the desire to directly control interactions between specific parties.

Nevertheless, many empirically observable networks can be characterized and classified according to a number of different network statistics, such as how densely connected they are, whether they are prone to clustering, etc. These structural properties, collectively referred to as a network's 'topology', influence how signals propagate through the network. Recognizing this, rather than propose a measure to optimize networks for opinion diversity, we instead study the impact of a network's topology on the resultant opinion diversity of the system.

To study the role of topology in opinion diversity, we begin with an agent-based model of a system of interacting agents. The majority of the classic models of opinion dynamics assume that opinion update rules are deterministic. As such, most models guarantee convergence to a consensus opinion under very reasonable conditions of connectedness [15]. Opinions in the real world, both at the individual and societal level, are not governed deterministically by social influence nor are they static [16]. Therefore, we augment two well-established social influence models to incorporate random opinion fluctuations and examine various network configurations to ask what macroscopic-level attributes of social networks promote/hinder diversity in opinions of agents within a social network.

In this paper, we make the following contributions:

1. We take two well-established benchmark models, namely the DeGroot and Friedkin–Johnsen (FJ) models and adapt them to a noisy framework.
2. We present a testable measure of expected network *opinion diversity* of the proposed models. We derive an analytical expression for it that can be computed for an arbitrary network from the eigenvalue sequence of its adjacency matrix.
3. We show the impact that network density, clustering and community structure have on opinion diversity. We also demonstrate how, depending on how the underlying distribution is defined, random fluctuations can capture both endogenous or exogenous factors that contribute to opinion formation.
4. We empirically validate the predictions of the noisy DeGroot model by testing the extent to which it captures variations in opinions in online news data.

4000

The remainder of the paper is structured as follows: §2 provides background and discusses relevant recent advances in noisy opinion dynamics models and opinion diversity. Section 3 outlines the classical DeGoot and FJ models of opinion dynamics and then presents two new models, a *noisy Degroot model* and a *noisy FJ model* that extend them. Here we also introduce a measure of opinion diversity and derive an analytical expression for it from these models. In §4, we test that our measure of opinion diversity is consistent with results on synthetic data, while §5 studies the effect that aggregate levels of susceptibility—a parameter of the FJ model that captures an agent's openness to their neighbours' opinions—have on the level of opinion diversity. Section 6 shows how different network characteristics effect opinion diversity, and in §7, we demonstrate how the addition of stochastic opinion fluctuation can be used to capture people's innate need to distinguish themselves from the consensus and the effect that this may have on the distribution of subsequent opinions. Finally, in §8, we empirically test our model's predictions of opinion diversity across online news media networks before providing a general discussion and concluding remarks in §9.

*Notation*. Throughout this paper, we will use the following notation: Let $G(V, E)$ be a weighted network with $|V| = N$ nodes and $|E| = kN/2$ edges, where $k$ is the average degree. Let $A$ be the row-stochastic adjacency matrix of $G(V, E)$ such that the influence that the $i$th node has on the $j$th node is given by $A_{i,j}$. $A$ is referred to as the *trust matrix* since the value of $A_{i,j} = 1/k_i$ can be interpreted as level of trust between the $i$th and $j$th nodes (in practice we use $A_{i,j} = 1/k_i(1 + \eta)$ where $\eta$ is a small constant to ensure that the opinions follow a stationary process). At a given time, $t$, the $i$th node's opinion is given by a scalar value $y_{i,t}$. The vector of opinions spanning all $N$ nodes at time $t$ is $\mathbf{y}_t$.

# 2. Related work

## 2.1. Noisy opinion dynamics

Many proposed approaches to opinion dynamics modelling rely on agent-based models, where each individual is represented by an agent and the individual's opinion is represented by a real value. Many models have been proposed and each tries to capture one or more of the factors that influence how opinions evolve over time. The archetypal DeGroot model [17], for instance, captures the property of 'assimilation'; the tendency of people to adjust their opinions towards those with whom they interact. Extensions to this approach include the FJ model [18], which accounts for a person's propensity to cling to prior beliefs. Mäs *et al.* [19] incorporate the trade-off between assimilation and differentiation, while Golub and Jackson [2] capture homophily.

Most of the routinely analysed models begin with the assumption that any randomness in the system is in the initial conditions and that the evolution of the system itself is otherwise considered to be wholly deterministic. As long as the system is strongly connected, interactions lead to one of two final states. Either there is complete consensus, in which everyone tends towards the same terminal opinion, or there is *weak diversity*, in which everyone adopts one of a discrete number of terminal opinions [20,21]. These models do not capture individual free will [21], nor do they account for exogenous factors unrelated to social influence.

Previous attempts to incorporate exogenous noise focused on so-called bounded-confidence models which assume that a social network's trust matrix updates over time as agents add edges with other agents who share similar opinions and drop edges with those whose opinions are dissimilar. Pineda *et al.* [21], for instance, propose a model in which agents only listen to their neighbours with a given probability; otherwise, they simply update their opinions by following a uniform distribution. Zhao *et al.* [22] extend the bounded-confidence model to study leader-follower relationships with Gaussian environmental noise. Though not strictly a bounded-confidence approach, Mäs *et al.* [19] introduce a model that captures the natural internal struggle between wanting to conform with societal views and simultaneously striving for uniqueness. This is a factor that we also address in our model. In [19], the authors capture the emergence of self-organizing clusters without the fragmentation seen in other noisy models. Bounded confidence models are useful because they capture homophily. However, our research is focused on the effect of network topology on diversity of opinions, which is difficult to study in bounded-confidence models because the network topology itself does not evolve smoothly in such models.

Our line of research bears similarity with work from control theory, where many of the problems of interest are similar. Xiao *et al.* [23] use an averaging consensus model with adaptive noise to examine the *least-mean-squared consensus* problem of determining, given a particular graph with known nodes and edges, what edge weight minimizes steady-state mean-square deviation.

## 2.2. Quantifying discord

There is currently no universally accepted method to capture discord (i.e. the lack of global consensus) in a social network, though several measures have been proposed for different contexts. Mackin & Patterson [13] put forth a diversity index measure inspired by ecology and based on calculating probabilities that random individuals belong to a particular group. Their measure relies on setting a predefined threshold to distinguish boundaries between groups. Matakos *et al.*, [24] propose a measure of *polarization* of opinions in a system as the $\ell_2$-norm of the system's equilibrium vector, while Musco *et al.* [14] augment this measure to include *disagreement*, taken to be the squared difference between opinions of neighbouring nodes. Chen *et al.* [25] compare numerous measures that capture some form of 'conflict' risk, including polarization and diversity, and provide average- and worst-case estimates of these measures.

# 3. Noisy opinion dynamics model

We begin by reviewing the classical DeGroot and FJ models of opinion dynamics, and then extend them to a noisy domain.

The DeGroot model [17] is a simple mechanism of opinion propagation that assumes (i) that every individual in a population has an opinion, and (ii) that everyone's opinion updates synchronously as a weighted average of their opinion plus the opinions of their friends. In other words, for a population of $N$ social *agents* (or actors), indexed 1 through $N$, who interact according to graph $\mathcal{G}$, each individual, $i$, has a scalar opinion $y_{i,t} \in \mathbb{R}$ at time $t$. This can be understood as a sentiment score representing how strongly an individual feels about a particular issue or how much they like/dislike something. The agents update their opinions synchronously from time $t$ to time $t+1$ via a weighted average of their own opinion at time $t$ and the opinions of their neighbours in $\mathcal{G}$. Mathematically this can be expressed as

$$y_{i,t} = \sum_{j=1}^{N} A_{j,i} y_{j,t-1}, \tag{3.1}$$

where $A$ is the trust matrix, as defined in §1. The above equation takes the same form of a discrete-time Markov chain. However, let us remark that the dynamics described by equation (3.1) are fully deterministic. The entry $A_{j,i}$ represents the level of influence that actor $j$ has on actor $i$. Using a column vector $\boldsymbol{y}_t = (y_{1,t} \cdots y_{N,t})^T$ the system can be written in compact notation as

$$\boldsymbol{y}_t = A \boldsymbol{y}_{t-1}. \tag{3.2}$$

The FJ [18] model is an extension to the standard DeGroot model that introduces an additional term $S$ that captures the trade-off between agents' susceptibility to social influence versus their stubbornness to cling on to their initial beliefs (prejudices). In the FJ model, agents' opinions evolve according to

$$\boldsymbol{y}_t = S A \boldsymbol{y}_{t-1} + (\mathbb{1} - S)\boldsymbol{\rho}, \tag{3.3}$$

where $S = \text{diag}(s_1, \ldots, s_N)$ is a diagonal matrix of susceptibilities where $s_i \in [0, 1]$ stands for the susceptibility of the $i$th agent and $\rho \in \mathbb{R}^N$ is a column vector of prejudices. For the purpose of analytical tractability, we make the simplifying assumption $S = s\mathbb{1}$. Note that if every individual has maximum susceptibility (i.e. $s = 1$), then we recover the DeGroot model of equation (3.2).

We propose two new models that extend those in equations (3.2) and (3.3) to allow for random fluctuations in agents' opinion. The first is a *noisy DeGroot model*, given as

$$\boldsymbol{y}_t = A \boldsymbol{y}_{t-1} + \boldsymbol{\epsilon}_t(\boldsymbol{y}_{t-1}, A), \tag{3.4}$$

where $\boldsymbol{\epsilon}_t(\boldsymbol{y}_{t-1}, A)$ is a random column vector. $\boldsymbol{\epsilon}_t$ can be thought of as the aggregation of the various idiosyncratic factors that affect people's opinions. Assuming such factors to be independent and drawn from distributions with finite variance, a central limit theorem argument allows us to assume $\boldsymbol{\epsilon}_t$ to be a Gaussian vector with zero mean and variance $\sigma^2\mathbb{1}$. However, in §7, we will consider alternative realizations of $\boldsymbol{\epsilon}_t$ to take into account scenarios where agents are exposed to non-idiosyncratic noise factors.

The second model is a *noisy FJ model* as

$$\boldsymbol{y}_t = S A \boldsymbol{y}_{t-1} + (1 - S)\rho + \boldsymbol{\epsilon}_t(\boldsymbol{y}_{t-1}, A). \tag{3.5}$$

There is subtle but important distinction between models with and without noise. In equations (3.2) and (3.3), the adjacency matrix is stochastic, i.e. $\sum_{j=1}^{N} A_{i,j} = 1$. By contrast, for the dynamics of equations (3.4) and (3.5) to be stable and stationary, the adjacency matrix must be strictly substochastic, i.e. $\sum_{j=1}^{N} A_{i,j} < 1$.

## 3.1. Opinion diversity

Noisy models of opinion dynamics will not converge to a stable consensus, but will instead converge on a stable distribution of opinions. To study the factors that impact this distribution, we propose a measure which we call the *opinion diversity*, as the expected squared deviation, $d$, of an arbitrary agent's opinion from the population mean $\bar{y}_t$ at time $t$, i.e.

$$d_t = \frac{1}{N}\sum_{i=1}^{N}\frac{(y_{i,t} - \bar{y}_t)^2}{N}. \tag{3.6}$$

For a stable system, we can equivalently denote the long-run expected squared deviation as

$$d = \frac{1}{N}\mathrm{Tr}[\mathrm{Cov}[\boldsymbol{y}]]. \tag{3.7}$$

In the steady state, $d$ measures how well the trust matrix $A$ enforces consensus despite the additive errors introduced by each node at each step [23].

## 3.2. Noisy DeGroot model

By expanding the model in equation (3.4) as an infinite series and inserting it into the above equation, we can write it as

$$d = \frac{1}{N}\mathrm{Tr}\left[\mathrm{Cov}\left[\sum_{k=0}^{\infty}A^k\boldsymbol{\epsilon}_{t-k}\right]\right]. \tag{3.8}$$

Then, for i.i.d. $\boldsymbol{\epsilon}$ with zero-mean and variance $\sigma^2$, the distribution of opinions will, as a result of the central limit theorem, also follow a Gaussian distribution. This results in an expression for the opinion diversity in terms of the spectrum of the adjacency matrix as

$$d = \frac{\sigma^2}{N}\sum_{i=1}^{N}\frac{1}{1-\lambda_i^2}, \tag{3.9}$$

where $\lambda_i$ is the $i$th eigenvalue of $A$ such that $\lambda_1 \geq \lambda_2 \geq \lambda_N$. For a detailed derivation of equation (3.9), see appendix A.

## 3.3. Noisy FJ model

If all agents in the system share the same susceptibility, i.e. $S = s\mathbb{1}$ and i.i.d. prejudices with variance $\xi^2$, then using a similar method we can obtain a closed-form expression for opinion diversity of the FJ model as

$$d = \frac{1}{N}(\sigma^2 + (1-s)^2\xi^2)\sum_{i=1}^{N}\frac{1}{1-s^2\lambda_i^2}. \tag{3.10}$$

If every actor is perfectly susceptible, (i.e. $s = 1$), then we retrieve the expression for the DeGroot model in equation (3.9). If, by contrast, everybody is completely zealous (i.e. $s = 0$), then $d$ becomes completely independent of the graph and the opinion diversity is just the sum of the variances of the initial conditions $\sigma^2$ and of the prejudices $\xi^2$, i.e.

$$d_{|s=0} = \sigma^2 + \xi^2. \tag{3.11}$$

It is worth highlighting that the expressions in equations (3.9) and (3.10) are fully general, and hold regardless of the specific network topology. Therefore, whenever two networks' adjacency matrices are isospectral, they will result in identical opinion distributions.

# 4. Results

Equation (3.9) provides an expression for the expected level of opinion diversity of a system that follows the noisy DeGroot model. We begin by validating that the expression accurately reflects the realized distribution of opinions (the code for this study is available https://github.com/samstern/noisy-opinion-dynamics). We generate 900 Erdös–Rényi random networks, each of which has 100 nodes.

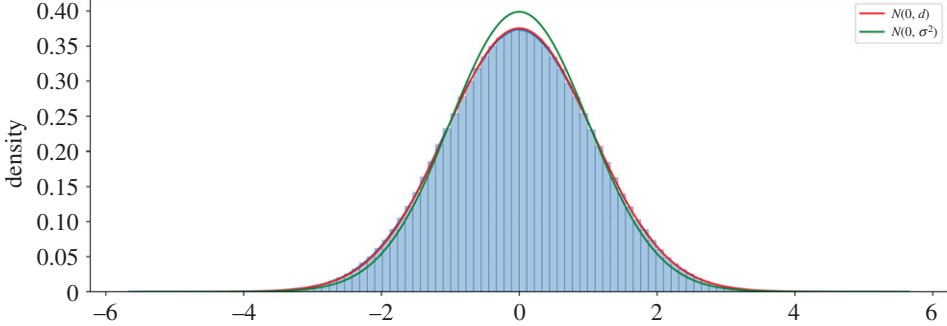

**Figure 1.** Steady-state distribution of opinions. $N(0, d)$ is a better fit of the data than $N(0, \sigma^2)$.

The networks vary in how connected they are, with 100 graphs having connectivity $p = 0.1$ (with connectivity being the probability of an edge existing between any pair of nodes), 100 graphs having $p = 0.2$, and so on, all the way to $p = 0.9$. The networks are all connected, and each node has a self-edge. The edges are uniformly weighted such that the adjacency matrices are all row-substochastic. Specifically, the edges of the $i$th node are set to $1/k_i(1 + \eta)$ where $0 < \eta \ll 1$ so that the rows are almost row-stochastic but the opinions will still follow a stationary process. Finally, for each network, we run 100 simulations of the process in equation (3.9) for 500 time steps each.

If the expression in equation (3.9) is correct, then the resultant opinions should follow a Gaussian distribution with mean 0 and variance $d$. Figure 1 shows the realized distribution of opinion values based on the simulation results. It also shows the expected distribution $N(0, d)$ as well as a benchmark model of $N(0, \sigma^2)$ for comparison. Visually, it is clear that the expected distribution of opinions closely follows the realized values, denoting a better fit than the benchmark model.

We can quantitatively validate goodness of fit for our estimate of the opinion distribution using a one-sample Kolmogorov–Smirnov (KS) test. The KS test is a non-parametric test of the empirical distribution, $F(x)$, of an observed random variable against a given distribution, $G(x)$. Under the test's null hypothesis, the two distributions are identical, $F(x) = G(x)$. In our case, $F(x)$ is the agents' set of terminal opinions and $G(x)$ is the normal distribution $N(0, d)$. We also perform the test against a benchmark model, $G_{\sigma^2} = N(0, \sigma^2)$. We perform two KS tests for each of the networks, one for $F(x) = G(x)$ and the other for $F(x) = G_{\sigma^2}(x)$, and use the Benjamini–Hochberg error correction method for multiple hypothesis testing [26]. Of the 900 trials, 89.7% reject the null hypothesis that the opinions follow the distribution $G_{\sigma^2}(x) = N(0, \sigma^2)$ at the 95% significance level. By contrast, only 2% of cases reject the null hypothesis that the opinions follow the distribution $G(x) = N(0, d)$. This demonstrates that for the overwhelming majority of cases, $d$ is an accurate predictor of the expected variance in opinions. The expression in equation (3.9) applies to undirected graphs. Strictly speaking, because our networks are not regular, by weighting the edges to make them row-stochastic, they are actuality directed. Nonetheless, the above test indicates that equation (3.9) is still a reliable indicator of the expected opinion diversity. While, to the best of our knowledge, there is no precise analytical expression for $d$ in the case of a directed network, we do derive an expected upper-bound in appendix A.

Table 1 breaks down the results according to how connected the network is based on its connectivity $p$. As shown in the table, $d$ is an accurate predictor of opinion diversity for the majority of network configurations, although this weakens as networks become more sparse. We therefore explore how disparate network configurations lead to differences in opinion diversity in subsequent sections of this paper.

# 5. Impact of susceptibility

The majority of people do not fully adopt the opinions of their peers on any given topic. This is due to a multitude of factors, including reliance on prior opinions and inputs from outside the network. Private belief models, such as the FJ model, suggest that people are not necessarily wholly susceptible to social pressures. Even though social planners cannot directly control peoples' innate levels of susceptibility, understanding the effect that varying susceptibility has on behaviour can give more accurate insight into the levels of opinion diversity and subsequently be used to inform policy.

In §3, we derived an expression for the expected opinion diversity of a system in which all actors have both a prior belief $\boldsymbol{\rho} \sim N(0, \xi^2 \mathbb{1})$ and susceptibility $S = s\mathbb{1}$. We now repeat the steps described above to

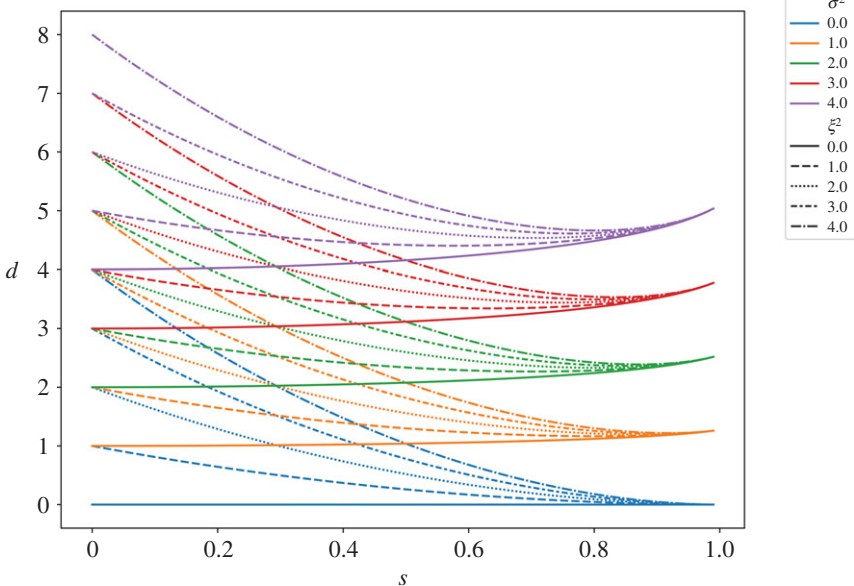

**Figure 2.** How opinion diversity varies with the susceptibility for different configurations of noise.

**Table 1.** The proportion of KS tests for which the null hypothesis is rejected.

| $p$ | $N(0, \sigma^2)$ | $N(0, d)$ |
|---|---|---|
| 0.1 | 100.0 | 8.0 |
| 0.2 | 100.0 | 4.0 |
| 0.3 | 100.0 | 4.0 |
| 0.4 | 98.0 | 0.0 |
| 0.5 | 90.0 | 0.0 |
| 0.6 | 82.0 | 0.0 |
| 0.7 | 78.0 | 0.0 |
| 0.8 | 86.0 | 0.0 |
| 0.9 | 74.0 | 0.0 |
| total (%) | 89.7 | 2 |

test whether our expression for opinion diversity is consistent with observed mean-squared opinion deviations for systems that follow the noisy FJ model. For a fixed $p$, we first generate 20 random graphs, each with 100 nodes. Then, for each graph and for $s$ ranging from 0 to 1 in increments of 0.2, we run 20 simulations and then perform the KS test. As before, with only 5% of cases where the null hypothesis is rejected, there is no compelling evidence that the realized distribution of opinions differs from the expected distribution.

Figure 2 shows how the opinion diversity varies with the susceptibility of the population. When actors are completely unsusceptible to their peers and are thus only influenced by exogenous noise ($s = 0$), we observe that the opinion diversity of the system is just the sum of the variance of the exogenous noise and the variance in prejudices between actors (see equation (3.11)). At the other extreme, when all actors are fully susceptible to each other's opinions ($s = 1$) the distribution of prejudices has no impact at all. The variance of the noise, $\sigma^2$, broadly has the effect of raising or lowering the level of opinion diversity, while the variance in the prejudices, $\xi^2$, broadly dictates the slope. It is noteworthy that the impact of susceptibility on opinion diversity is non-monotonic. That is to say, for any single realization of a social network, there may be more than one level of susceptibility that leads to the same behaviour.

# 6. Impact of topology on opinion diversity

In our model, opinion diversity is dependent on four factors: the amount of noise in the system, the number of actors, their susceptibility to social influence and the structure of the network. Having explored how susceptibility impacts opinion diversity, we proceed to study how the structure of a network impacts the resultant diversity. We approach this by looking at the macroscopic characteristics exhibited by real-world social networks, specifically connectivity, clustering, and community structure, to understand how differences in network topology can impact the level of agreement between agents.

## 6.1. Connectivity

The connectivity of real-world social networks varies depending on the domain. Twitter topic networks, for example, are often more densely connected on politically sensitive topics than on topics that are apolitical [11]. In an Erdös–Rényi network, the connectivity $p$ is the probability of any two agents sharing an edge. By adjusting $p$, we can control how densely or sparsely the network is connected, and thereby test its impact on a sample population's opinion diversity. Using the results from the previous simulation, we compare the distributions of opinion diversity across the various randomly generated networks and then group them according to how connected they are. Figure 3 shows the distributions of the opinion diversity for each group of networks. The figure highlights two features; the first showing that opinion diversity clearly decreases as a network becomes more densely connected. The relationship between connectivity and opinion diversity is not linear, but rather logarithmic; an increase in $p$ by 1% on average leads to a 0.03% decrease in opinion diversity. The second feature that can be seen is the variance within each of these groups also decreases as networks become more densely connected, which explains why, in the KS test in §4, sparse networks were more likely to reject the null hypothesis. This is probably due to the fact that, in the case of sparse networks, a comparatively much larger number of network configurations leads to the same connectivity.

The differences in the dynamics for varying macroscopic network characteristics can be attributed to differences in the eigenvalue sequence. The eigenvalues are all real, and strictly less than 1. Each eigenvalue's 'contribution' to the diversity is proportional to $1/1 - \lambda_i^2$. If the system is fully connected, then the largest eigenvalue, $\lambda_1$, will have an algebraic multiplicity of 1 and the remaining eigenvalues will be 0. If the network is entirely disconnected such that the actors are fully independent of one another, then the algebraic multiplicity of $\lambda_1$ will be $N$.

Figure 4 illustrates why we see distinctions in diversity for varying degrees of connectivity. The eigenvalue index, $i$, ranked by its value, $\lambda_i$, is on the $x$-axis, while the marginal contribution of $\lambda_i$ to $d$ is on the $y$-axis. Each curve represents one of the randomly generated networks. The long-run opinion diversity for each network is directly proportional to the integral of its respective curve in figure 4. What we can see is that the integral increases as networks become sparser.

## 6.2. Clustering

Moving forward, we now consider the effect that varying the level of clustering has on opinion diversity for a fixed average degree. It is well known that in many real-world social networks, the probability of a social tie existing between any pair of individuals is not independent of other social ties, instead, it is dependent on whether they have shared neighbours. Real social networks exhibit greater triadic closure (clustering): two nodes that share a common neighbour have an increased probability of being neighbours themselves. Small-world networks, which have both high triadic closure as well as short average path lengths, are of particular interest. This is not only because many real-world social networks exhibit small-world properties, but also because signals propagate particularly efficiently through small-world networks relative to other network structures. For a $k$-regular graph, increasing the average degree of a network (which we have already established has the effect of decreasing opinion diversity) is directly linked with a decreased average path length between nodes.

The Watts–Strogatz algorithm [27] is a popular method for generating random graphs that exhibit varying degrees of the small-world characteristic. The method arranges $N$ nodes in a circular lattice, where each node shares an edge with its $k$ nearest neighbours in the lattice, so that the degree of each node is $p = k/N$. Each of the edges are then randomly rewired with probability $q$.

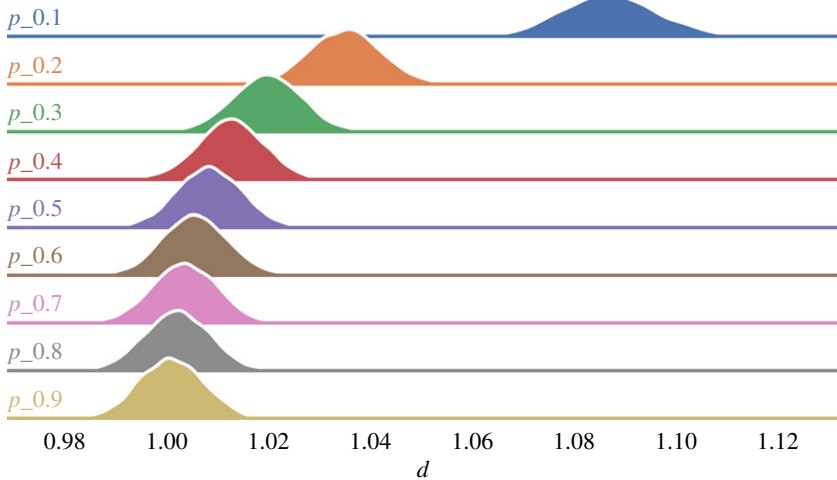

**Figure 3.** Distribution of opinion diversity, $d$, grouped by network connectivity, $p$.

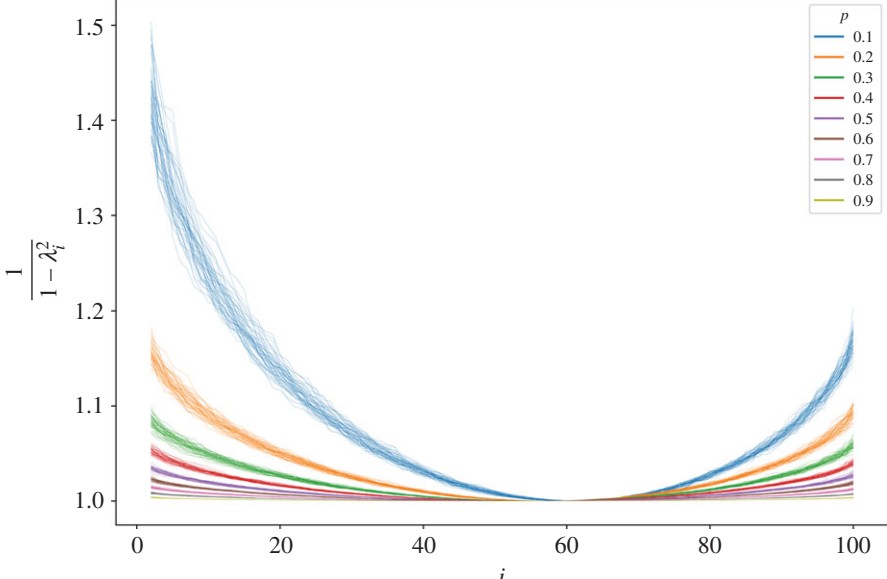

**Figure 4.** The marginal contribution of $\lambda_i$ to $d$ for each network.

By varying $q$, we can control how clustered the network is. If $q = 0$, then there is no rewiring and the network has maximal clustering as well as maximal average path length between nodes. As $q$ increases, the number of closed triplets decreases. This, in turn, decreases clustering while increasing the number of long-range connections between clusters, thus decreasing the average path length. Finally, when $q = 1$, we obtain networks with similar properties to Erdös–Rényi random graphs.

To understand the impact that clustering has on opinion diversity, we run a similar experiment to that described above. However, rather than using Erdös–Rényi-generated edges between nodes, the graphs are constructed using the Watts–Strogatz method. We then compare the distributions of realized opinion diversity of our graphs for varying $q$.

As shown in figure 5, the impact of clustering on variations in opinion diversity is markedly less than the impact of connectivity. That notwithstanding, it is clear when looking at a fixed average degree, that as clustering increases, so does opinion diversity. It is likely that this relationship is due to differences in average path length, since networks with lower rewiring probability hold fewer long-range connections. This translates to signals propagating less efficiently from one end of the network to the other, compared with networks with a greater proportion of long-range connections.

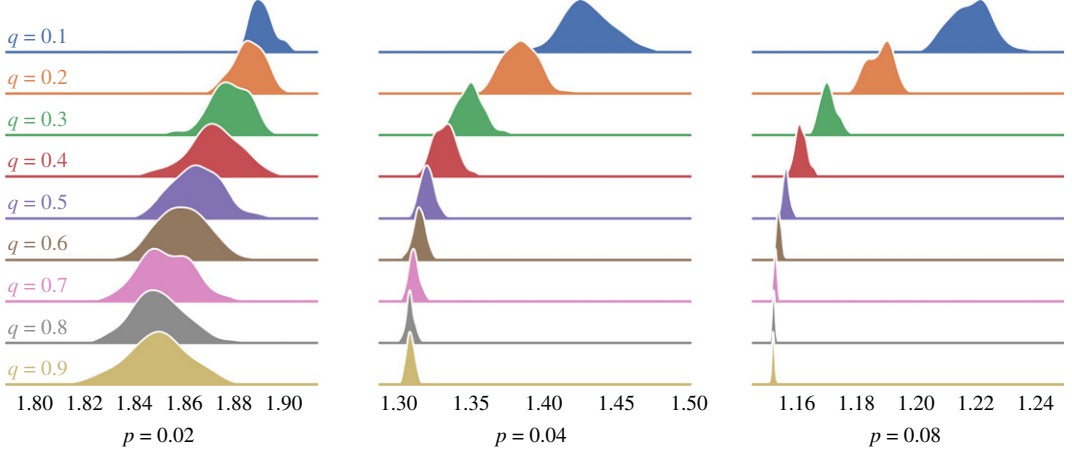

**Figure 5.** Distribution of opinion diversity grouped by $q$ (rows) and $p$ (columns).

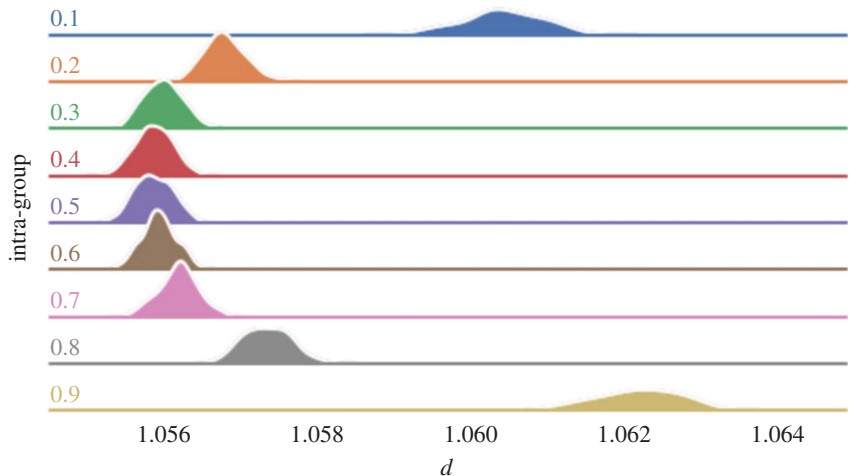

**Figure 6.** Distribution of opinion diversity for networks with average degree $k = 0.5N$.

## 6.3. Communities

The final element of network topology that we examine is the 'strength of community ties'. Here, we are interested in answering the question: does the strength of community ties affect opinion diversity? Communities in networks are groups of nodes that are more densely connected to one another than they are to nodes that fall outside the group. They are pervasive in many online social networks such as Twitter, for example, where it has been well-documented that people form communities that are strongly aligned with their political ideologies [28]. We test this by comparing the opinion diversity of a collection of randomly generated graphs with varying degrees of community strength. The stochastic block model [29] is widely employed as a canonical model to study the trade-off between inter-group and intra-group connections in social networks. For a system with $m = 2$ communities, the stochastic block model for generating networks takes two parameters. The first parameter is a vector $\boldsymbol{n} \in \mathbb{N}^m$, where $n_i$ is the number of nodes in the $i$th group. The second is a symmetric $m \times m$ matrix, $P$, where $P_{i,j}$ is the probability that an arbitrary node from group $i$ shares an edge with an arbitrary node from group $j$. By adjusting the relative values between the diagonal and off-diagonal elements of $P$, we generate random graphs with varying degrees of inter- and intra-group connectivity. We can also adjust the average degree of the network through sum of the rows. For an intra-group connectivity $P_{i,i} = \pi$, the inter-group connectivity is simply $2k/N - \pi$.

Figure 6 shows how opinion diversity changes with the intra-group connectivity for a fixed average degree. Opinion diversity is lowest when $\pi \approx k/N$ (in other words when there is no clear distinction between groups and therefore no communities). This is consistent with the previous set of results where we established that opinion diversity decreases as networks tend towards Erdös–Rényi random graphs. It then increases again, either when the inter-group connectivity is much larger than the intra-group connectivity or vice versa. So-called heterophilic networks, in which inter-group connectivity exceeds intra-group, are uncommon in

networks that govern the social patterns of relationships and/or group formations. It may still be observed, though, across groups formed by individuals with complementary skill sets, such as collaborative work teams [30]. However, the other extreme, where intra-group connectivity exceeds inter-group connectivity is ubiquitous and is often referred to as the 'echo-chamber' effect, particularly when group divisions coincide with political ideology [31,32]. Our results therefore add to the growing body of the literature that argues that high modularity and closed communities can contribute to heightened discord.

# 7. Adaptive noise

Up to this point, we have assumed that the stochastic component, $\epsilon(y_t, A)$, of the model represents the aggregation of any exogenous factors other than social influence that sway opinions. Therefore, we assumed it to be i.i.d. Gaussian (i.e. $\epsilon(y_t, A) \sim N(0, \sigma^2)$) based on a central limit theorem argument. However, the stochastic component can also capture endogenous sources of non-deterministic behaviour. There is evidence, in fact, suggesting that, in addition to the social influence that causes people's opinions to gravitate towards one another's, people also often 'strive for uniqueness' [19,33,34]. Optimal distinctiveness theory [35] posits that humans experience competing needs of *assimilation* (fitting in) and *differentiation* (standing out). While individuals are inclined to conform by their social environment, they also exhibit a desire to increase their uniqueness when too many other members of the society hold similar opinions.

We continue by addressing the question of how the trade-off between assimilation and differentiation can be captured within our model and, furthermore, how this might impact the distribution of opinion diversity. Assimilation is captured naturally by our models through the interaction weights of the trust matrix, though differentiation is not. However, as pointed out by Mäs *et al.* [19], we can account for peoples' desire to be unique by setting the stochastic fluctuations of an individual's opinion to be proportional to how unique their opinion is. With this in mind, we propose two variants of the noisy DeGroot model: a *global uniqueness* (GU) model, which assumes that individuals globally strive to be unique from the remainder of the population, and a *local uniqueness* (LU) model, that models individuals as striving to be unique from their neighbours.

The GU model is distinguished by

$$\epsilon_{i,t} \sim N(0, \sigma^2 \, e^{-\beta(y_{i,t-1} - \bar{y}_{t-1})^2}), \tag{7.1}$$

whereas the LU model is distinguished by

$$\epsilon_{i,t} \sim N\left(0, \sigma^2 \sum_{j=1}^{N} A_{i,j} \, e^{-\beta(y_{i,t-1} - y_{j,t-1})^2}\right), \tag{7.2}$$

where $\beta \geq 0$ is a constant that controls how strongly an individual's desire to be unique decays as their opinion deviates from the consensus. When $\beta = 0$ we get to the non-adaptive noisy DeGroot model, in other words the case where noise is i.i.d. Gaussian. This has the nice property that the non-adaptive noisy DeGroot model can be seen as an upper bound. At $\beta \longrightarrow \infty$, people have no desire to be unique at all and we retrieve the deterministic DeGroot model.

The GU model assumes that every actor has knowledge about how their opinion differs from the global mean. They continue to be influenced by their peers, however, the closer their opinion tends towards the mean, the greater the probability that their opinion will shift dramatically away from the mean. The GU model captures situations where polling provides knowledge of average opinions in a population and in which people are incentivized to stand out or be in some way contrarian and unique.

By contrast, the LU model assumes that actors only know the opinions of their immediate neighbours and not the global average. It captures the property that agents assimilate to their neighbours while at the same time desire to be recognized as individuals in their respective social circles.

Unlike for the non-adaptive case, there is no simple closed-form expression for the expected opinion diversity. To understand how opinion diversity differs under the varying models, we must therefore rely on simulation results. Using the random networks generated in the experiment in §6.1, we once again run 100 simulations of both the LU and GU model for each network and compare the distribution of opinion diversity for differing levels of $\beta$ and $p$.

Figure 7 shows, for fixed $\beta$, how opinion diversity changes with the average degree of a network. The realized opinion diversity under the LU model is consistently lower than that of the GU model. Just as with the non-adaptive model, opinion diversity decreases as networks become more densely connected for the LU model. A noticeable difference between the GU and the other models is that for higher $\beta$,

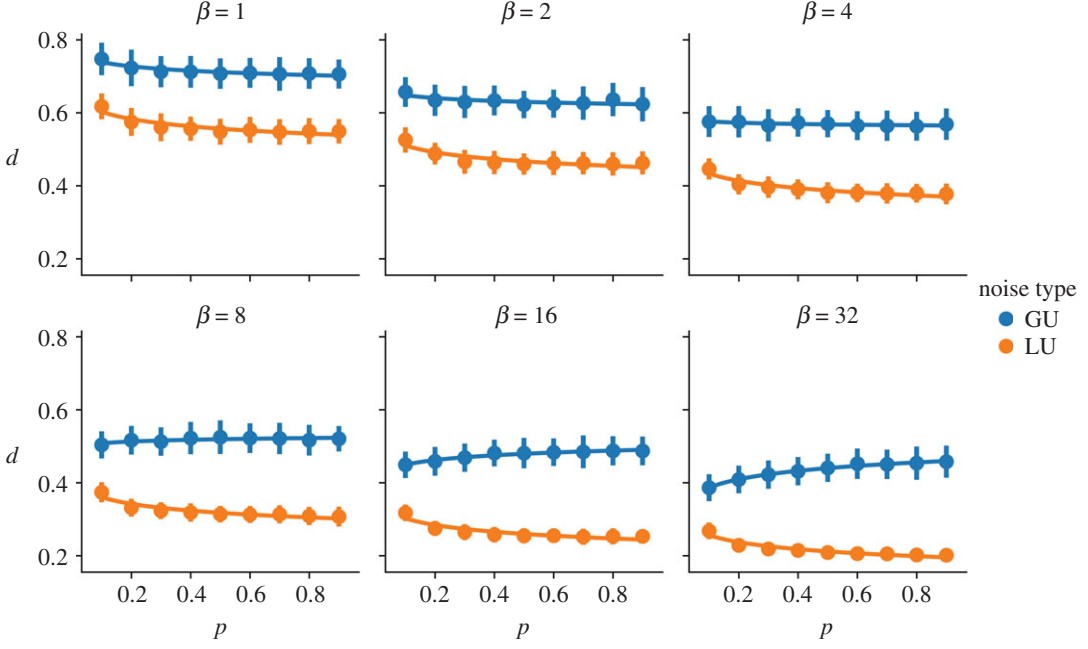

**Figure 7.** How opinion diversity varies with connectivity for the GU and LU model depending on $\beta$.

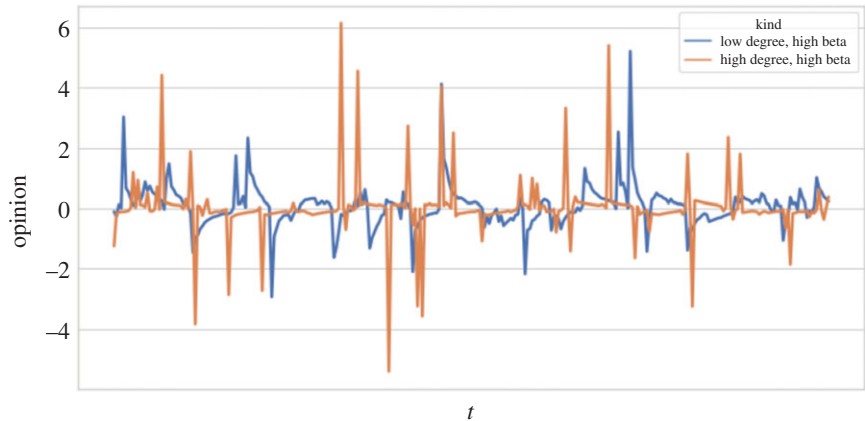

**Figure 8.** Example highlighting why $d$ increases with $p$ when $\beta \gg 0$.

opinion diversity increases, rather than decreases, with average degree. Figure 8 gives us intuition as to why this reversal occurs. It is an extreme example, for illustrative purposes, showing the opinion evolution of a node from two simulations, one where the node sits within a network with very high average degree and one where the node sits in a network with very low average degree. In both instances $\beta = 100$, $\sigma^2 = 1$ and $\mu = 0$.

At higher $\beta$, individuals have less desire to be unique, and the system becomes less random (indeed equation (7.1) is inspired by the Boltzmann distribution from statistical mechanics, with $\beta$ playing the role of its inverse temperature). When an individual's opinion is very close to the consensus, the model assumes that they feel a strong need to differentiate themselves, regardless of $\beta$. However, $\beta$ controls how quickly this desire tails off as an individual's opinion moves away from the consensus. When $\beta$ is large, the desire to be unique is not felt by an individual until their opinions are close to the population mean, at which point they then experience a sudden and strong desire to be unique. The resultant distribution of opinions is therefore more prone to outliers. This is experienced less in sparse networks, which advance towards consensus more slowly. For opinion dynamics models that use adaptive noise to capture the trade-off between assimilation and differentiation, it may be more appropriate to consider alternative robust measures of dispersion, such as median absolute deviation. In the example in figure 8, the mean-square deviations of the dense and sparse networks are 0.55 and 0.60, respectively, whereas their median absolute deviations are 0.17 and 0.37.

# 8. Empirical evaluation

In this section, we validate the noisy DeGroot model of opinion dynamics against an empirical dataset of opinions expressed in the news media. Opinion diversity within the media is a particularly important application domain of opinion dynamics because the media plays a central role in influencing the information that the public attends to. Diversity in media reporting is an important factor in promoting the reception of diverse ideas, especially for individuals with lower levels of education [36]. In a healthy society, it is important for the media to present a stable, balanced and diverse set of perspectives to the public [8,37,38].

We use the dataset collected by Stern *et al.* [39], which contains a collection of texts produced by a list of 97 prominent online news sources between May and December 2016. Following the method outlined in detail by Stern *et al.* [37], we use latent Dirichlet allocation (LDA) to identify 161 topics that are discussed throughout the eight-month period. For each of the topics, we then generate a sentiment index for each of the news sources that captures how the opinion of each of the news sources evolves over time for each of the topics. For each topic, we then construct networks, with nodes corresponding to news sources, and edges representing lead–lag relationships between the news sources. Specifically, for two agents $A$ and $B$, an edge is drawn from $A$ to $B$ if the conditional probability of observing $B(t)$ is dependent on $A(t - \Delta(t))$ for some time $t$. The relationship between pairs of news sources is measured using a Granger causality test [40], and we account for family-wise error rate by applying the false discovery rate correction, which we implement with the Benjamini–Hochberg procedure [26].

In so-called *intermedia influence* networks, such as the ones we test here, previous work has demonstrated that a simple linear model can capture the relationship between the variance in opinions across news sources and macro-level network properties, e.g. the average clustering coefficient and density [37]. Having demonstrated that in synthetic data these network properties are encoded within the spectrum of the graph, we proceed by testing whether the expected opinion diversity is an indicator of realized intermedia opinion diversity.

To test this, we consider a series of linear models that attempt to capture these effects and compare the models based on the significance of their coefficients. The dependent variable, $\log(y)$, is the log of the mean-square deviation of news sources opinions (we take the log because the realized opinions are lognormally distributed). Formally, for $k \in \{0, \ldots, K\}$, $y_k$ is defined as

$$y_k = \text{mean}\{\text{Var}[H_{t,\bullet,k}] \,|\, t \in \{0, \ldots, T\}\} \tag{8.1}$$

$$= \text{mean}\left\{ \frac{1}{N_k} \sum_i^{N_k} (H_{t,i,k} - \overline{H_{t,\bullet,k}})^2 \,|\, t \in \{0, \ldots, T\} \right\}, \tag{8.2}$$

where $N_k$ is the number of actors in the $k$th network and $H_{t,i,k}$ is the opinion of the $i$th actor in the $k$th network at time $t$. We train three linear regression models, each with weights learned using maximum-likelihood estimation. The first model is that proposed by Stern *et al.* [37], and is of the form, $\log(y) = \omega_0 + \omega_1 L + \omega_2 N + \omega_3 C + \omega_4 D$, where $L$ is the average shortest path length, $N$ is network size, $C$ is the average clustering coefficient, $D$ is the network density. The second regression is of the form $\log(y) = \omega_0 + \omega_1 d$, where $d$ is the opinion diversity calculated using equation (3.9). The third and final model includes all the independent variables in both of the previous two models. Since we cannot directly estimate $\sigma^2$, models $M_2$ and $M_3$ implicitly assume that it is constant across networks and is contained within the regression coefficient. This is a reasonable assumption since the nodes of the networks all come from the same population and only the edges are changed.

Table 2 compares the three linear models. From $M_2$, there is clear evidence that we can reject the null hypothesis that $y$ is independent of $d$ with 99% confidence. Furthermore, despite being a very simple model, $M_2$ is able to capture 19.2% of the opinion variance in the online news media, increasing to 25% in the $M_3$ model. While intermedia influence networks are only one of many possible empirical social networks, the fact that there is a strong relationship between the influence networks and the resultant dynamics validates that network topology has a measurable impact on the level of discord.

# 9. Conclusion

In this paper, we consider the question of measuring discord in social networks. Building on a popular model of opinion formation, we investigate the impact of both noise and network topology on discord. In this context, we propose an index of *opinion diversity* based on the expected mean-squared deviation of

**Table 2.** The regression coefficients of each of the three linear models. Model $M_1$ is $\log(y) = \omega_0 + \omega_1 L + \omega_2 N + \omega_3 C + \omega_4 D$, model $M_2$ is $\log(y) = \omega_0 + \omega_1 d$, model $M_3$ is $\log(y) = \omega_0 + \omega_1 L + \omega_2 N + \omega_3 C + \omega_4 D + \omega_5 d$, where $y$ is the opinion diversity, $L$ is the average shortest path length, $N$ is network size, $C$ is the average clustering coefficient, $D$ is the network density and $d$ is the expected opinion diversity. The values shown below pertain to the coefficients $\omega_i$ ($i = 0, \ldots, 5$) with $p$-values reported in brackets. ($^*p < 0.1$; $^{**}p < 0.05$; $^{***}p < 0.01$.)

|  | $M_1$ | $M_2$ | $M_3$ |
| --- | --- | --- | --- |
| intercept | −14.03 | −2.14 | −2.80 |
|  | (0.001***) | (0.001***) | (0.003***) |
| Avg. shortest path length ($L$) | −0.01 | – | −0.03 |
|  | (0.967) |  | (0.311) |
| network size ($N$) | 0.98 | – | −0.10 |
|  | (0.042*) |  | (0.431) |
| Avg. clustering coefficient ($C$) | 0.67 | – | 0.10 |
|  | (0.024**) |  | (0.027*) |
| density ($D$) | −0.95 | – | −0.03 |
|  | (0.006***) |  | (0.712) |
| Exp. opinion diversity ($d$) | – | 0.49 | 0.80 |
|  |  | (0.001***) | (0.016**) |
| $R^2$ | 0.223 | 0.192 | 0.250 |

scalar opinions. We derive an analytical expression for this index, which holds for any network and depends on its structural specificities through its adjacency matrix's eigenvalue sequence.

Having first validated that our index reliably captures diversity in synthetic data, we then examine the problem of identifying how network structure affects opinion diversity. We observe that there is a clear inverse relationship between the density of a network and the amount of diversity it generates. This relationship is the result of the trade-off between the assimilating forces that draw opinions to convergence on a consensus, and the random fluctuations of individuals' opinions that inhibit consensus. We further find that the more social ties exist between random individuals, the quicker each individual's opinion propagates, and the stronger the assimilating force becomes relative to random opinion fluctuations. We then consider the effect of clustering and community structure. In both cases, we find that opinion diversity decreases on average when networks become less structured. This finding is consistent with existing work, such as Musco *et al.* [14], that finds evidence suggesting that Erdös–Rényi graphs minimize discord in a system. Likewise, we reinforce findings that, at least in principle, breaking down barriers between isolated communities can help build social capital [41]. These results can act as heuristics on how to either increase or decrease discord in a community.

We also demonstrate how stochastic opinion dynamics models can be used to capture the trade-off between assimilation and differentiation, in addition to exogenous noise. Our results convey that if all individuals perturb their opinions in proportion to how much they conform, then the resultant behaviour is one in which people fluctuate between periods of relatively low and relatively high volatility. There is empirical evidence stemming from the field of social psychology supporting that people seek to establish and maintain a moderate level of uniqueness. In fact, the volatile behaviour seen in the GU model in particular is suggestive of attributes associated with people prone to believe in conspiracy theories [42]. Our model assumes that all individuals experience the need for uniqueness at the same level, $\beta$. However, any one person's desire to be unique is probably context- and time-dependent, and varies from individual to individual [43]. Therefore, we believe that potentially fruitful future research lies in extending our model to assign each individual a distinct $\beta$, in order to understand how individuals or communities with an exceptionally high/low need for uniqueness may impact the long-run distribution of opinions in a population.

Though weighted-averaging models such as those we employ have received substantial lines of validation in human-subject experiments (e.g. [44,45]), validating them in large-scale experiments has historically proven more difficult. This is mainly because of the obstacles researchers face when trying to acquire datasets that contain information on both the networks of social influence and on the temporal dynamics of the agents within them.

Fortunately, access to online forums and social media platforms has helped facilitate some big-data-driven empirical research, for example, studies confirming relationships between network modularity and opinion polarization [28]. It should be noted, however, that many of the platforms used in these studies, such as Twitter and Reddit, suffer from participation asymmetry, or the so-called 1% rule, where only 1% of Internet communities generate the content that is viewed by the remaining 99%.

Our model provides testable predictions about the relationships between the social structure of a population and the amount of disagreement it experiences. As such, our final contribution was to empirically validate our approach to opinion diversity on so-called *external* diversity within the media [46]. That is to say, we looked at the level of diversity across, rather than within, different news sources. Compared with previous studies on the same dataset, we were able to generate a 12% increase in the $R^2$ of models aimed at capturing external diversity. Our results augmented similar findings from data-driven studies geared toward understanding the impact of network structure on discord. Garimella *et al.* [47], for example found evidence that network structure can be used to identify controversial topics, while Guerra *et al.* [48] similarly showed that community boundaries, and modularity in particular, link with polarization of viewpoints on Twitter.

All of the analyses that we perform, both simulation-based and empirical, are carried out on homogeneous networks. Some of the network features we analyse, such as community structure and average degree, do not generalize well to heterogeneous networks, such as the Barabási–Albert model [49]. However, the analytical expressions in equations (3.9) and (3.10) are fully general and also apply to the heterogeneous case. As such, they provide a useful tool to quantify the impact of influencers (i.e. major hubs in social networks) on opinion diversity.

Our model captures some of the most well-documented factors contributing to opinion formation, including assimilation, stubbornness and differentiation. We acknowledge, however, that there are numerous other factors influencing opinion formulation that are not captured by our model. For example, the DeGroot and FJ models do not account for human homophily (the tendency for people to form ties with those who are similar to them and remove ties if they become dissimilar). Our work therefore opens other interesting questions, such as whether similar results could be observed if we were to use more complex models of opinion dynamics, or alternative measures of discord.

Data accessibility. The data and relevant code for this research work are stored on GitHub (https://github.com/samstern/noisy-opinion-dynamics) and have been archived within the Zenodo repository (https://doi.org/10.5281/zenodo.4560090). The code to simulate the models described in the paper has been uploaded to GitHub: https://github.com/samstern/noisy-opinion-dynamics.

Authors' contributions. S.S. performed all simulations and data analyses. Both authors designed the research and wrote the paper. Both authors gave final approval for publication and agree to be held accountable for the work performed therein.

Competing interests. We declare we have no competing interests.

Funding. G.L. acknowledges support from an EPSRC Early Career Fellowship in Digital Economy (grant no. EP/N006062/1).

Acknowledgements. We are thankful to Simone Righi and Orowa Sikder for their feedback on preliminary versions of this manuscript.

# Appendix A. Analytical expression for opinion diversity

We wish to obtain an expression for the expected deviation, $d_t$, of a signal $y_{i,t}$ from the population mean $\bar{y}_t$,

$$d_t = \sum_{i=1}^{N} \frac{(y_{i,t} - \bar{y}_t)^2}{N}. \tag{A 1}$$

This is equivalent to the trace of the covariance matrix of $y_t$

$$d_t = \frac{1}{N} \operatorname{Tr}[\operatorname{Var}[y_t]]. \tag{A 2}$$

In the long run, $\operatorname{Var}[y_t]$ can be written as a sum of errors $\epsilon$,

$$\operatorname{Var}[y_t] = \operatorname{Var}\left[\sum_{k=0}^{\infty} A^k \epsilon_{t-k}\right] \tag{A 3}$$

$$= \sum_{k=0}^{\infty} A^k \operatorname{Var}[\epsilon_{t-k}](A^T)^k, \tag{A 4}$$

where $A^T$ is the transpose of $A$.

Since the error terms, $\epsilon_t$, are i.i.d. Gaussian with variance $\sigma^2 \mathbb{1}$, we can write the above expression as, 

$$\mathrm{Var}[y_t] = \sigma^2 \sum_{k=0}^{\infty} A^k (A^T)^k. \tag{A 5}$$

The steady-state expression for $d$ is then,

$$d = \frac{1}{N} \mathrm{Tr} \left[ \sigma^2 \sum_{k=0}^{\infty} A^k (A^T)^k \right] \tag{A 6}$$

$$= \frac{\sigma^2}{N} \sum_{k=0}^{\infty} \mathrm{Tr}[A^k (A^T)^k]. \tag{A 7}$$

## A.1. Undirected case

First, we consider the case where the graph $\mathcal{G}$ is undirected, in other words where $A$ is a symmetric matrix. Using the fact that $A = A^T$, the expression for $d$ can then be written as,

$$d = \frac{\sigma^2}{N} \sum_{k=0}^{\infty} \mathrm{Tr}[A^{2k}]. \tag{A 8}$$

By exploiting the property that the trace of a matrix is the sum of its eigenvalues, we obtain the expression,

$$d = \frac{\sigma^2}{N} \sum_{k=0}^{\infty} \sum_{i=1}^{N} \lambda_i^{2k} \tag{A 9}$$

$$= \frac{\sigma^2}{N} \sum_{i=1}^{N} [1 + \lambda_i^2 + \lambda_i^4 + \cdots]. \tag{A 10}$$

Observing that this is a geometric series, we obtain the final expression,

$$d = \frac{\sigma^2}{N} \sum_{i=1}^{N} \frac{1}{1 - \lambda_i^2}. \tag{A 11}$$

## A.2. Directed case

When the network is directed, we do not have an analytical expression for the expected diversity, however, we can obtain an expected upper-bound.

Since $A$ is a square matrix, it can be written in the diagonalized form $A = V \Lambda V^{-1}$, where $\Lambda$ is a $N \times N$ diagonal matrix containing the eigenvalues of $A$, and $V$ is an $N \times N$ matrix where the $i$th column of $V$ contains the normalized right eigenvector of $A$ corresponding to the $i$th eigenvalue $\lambda_i$, i.e. $V = (v_1^T \; v_2^T \; \ldots \; v_N^T)$ where $A v_i^T = \lambda_i v_i^T$. Similarly, each of the rows of $V^{-1}$ contains a left eigenvector, $u_i$ of $A$, i.e. $V^{-1} = \begin{pmatrix} u_1 \\ u_2 \\ \ldots \\ u_N \end{pmatrix}$ where $u_i A = \lambda_i u_i$.

Inserting the diagonalized form of $A$ and $A^T$ into equation (A7), we get

$$d = \frac{\sigma^2}{N} \sum_{k=0}^{\infty} \mathrm{Tr}[(V \Lambda V^{-1})^k (V^{-1^T} \Lambda V^T)^k] \tag{A 12}$$

$$= \frac{\sigma^2}{N} \sum_{k=0}^{\infty} \mathrm{Tr}[V \Lambda^k V^{-1} V^{-1^T} \Lambda^k V^T]. \tag{A 13}$$

In the case of symmetric $A$, the eigenvectors form an orthonormal basis, which results in the inner term simplifying to $V^{-1} V^{-1^T} = \mathbb{1}$. When $A$ is not symmetric, then the eigenvectors are not guaranteed to be orthogonal. They are, however, still linearly independent of one another and of unit-length. We exploit this fact to obtain upper bounds on the elements of $V^{-1} V^{-1^T}$,

$$|(V^{-1} V^{-1^T})_{i,j}| \begin{cases} = 1, & \text{if } i = j \\ \leq 1, & \text{otherwise.} \end{cases} \tag{A 14}$$

Multiplying by the diagonal matrix $\Lambda^k$ yields,

$$|\Lambda^k (V^{-1} V^{-1^T})_{i,j}| \begin{cases} = \lambda_i^k, & \text{if } i = j \\ \leq \lambda_i^k, & \text{otherwise.} \end{cases} \tag{A 15}$$

The same result holds for $V^T V$.

Since we are ultimately only concerned with the trace, we consider only the diagonal elements. The element $i$, $i$ of the covariance matrix $\text{Var}[\boldsymbol{y}_t]$ is,

$$\text{Var}[\boldsymbol{y}_t]_{i,i} \leq \frac{\sigma^2}{N} \sum_{k=0}^{\infty} \lambda_i^k \sum_{j=1}^{N} \lambda_j^k. \tag{A 16}$$

Substituting this back into the expression for $d$ and by the same logic we employed in the undirected case above,

$$d \leq \frac{\sigma^2}{N} \sum_{k=0}^{\infty} \sum_{i=1}^{N} \sum_{j=0}^{N} \lambda_i^k \lambda_j^k \tag{A 17}$$

$$= \frac{\sigma^2}{N} \sum_{i=1}^{N} \sum_{j=1}^{N} [1 + \lambda_i \lambda_j + \lambda_i^2 \lambda_j^2 + \cdots] \tag{A 18}$$

$$= \frac{\sigma^2}{N} \sum_{i=1}^{N} \sum_{j=1}^{N} \frac{1}{1 - \lambda_i \lambda_j}. \tag{A 19}$$

Note that the above expression results in a real number, even if some of the eigenvalues are complex. This can be demonstrated by noting that the trace of $A$ is real. Therefore, any possible complex eigenvalue $\lambda_i$ must be accompanied by a complex conjugate eigenvalue $\lambda_i^*$, which ensures that the imaginary parts of the terms in the above sums cancel each other out.

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
