## [Peer Review File · Royal Society Open Science]

Review History

RSOS-201943.R0 (Original submission)

Review form: Reviewer 1

Is the manuscript scientifically sound in its present form?

Yes

Are the interpretations and conclusions justified by the results?

Yes

Is the language acceptable?

Yes

Do you have any ethical concerns with this paper?

No

Have you any concerns about statistical analyses in this paper?

No

Recommendation?

Accept with minor revision (please list in comments)

Comments to the Author(s)

The authors study models of opinion dynamics on social networks and aim to quantify the effects of noise on the outcome, as well as the impact of network topology. They focus on two rather well studied models -- 1) DeGroot, and 2) Friedkin-Johnsen, and introduce their noisy counterparts, which they study on Erdos-Renyi random networks. They define a useful quantity termed 'opinion diversity', for which they provide a general formula for arbitrary undirected graphs, and show that it is a quantity of interest by working with a real dataset, and improving upon the previous modelling attempt on this dataset, by incorporating opinion diversity in the model. Overall, I think the authors have considered an interesting problem, and have presented a nice analysis -- with useful analytical results, followed by numerical verification and an application to real data. Keeping in mind the aim and scope of the Royal Society Open Science journal, I am happy to recommend this paper titled "The Impact Of Noise And Topology On Opinion Dynamics In Social Networks" for publication provided that the authors address the comments, and fix some minor typos/citation errors as mentioned below:

1. Throughout the work, the authors point to the fact that so far, most models of opinion dynamics that have been studied have been deterministic in nature (e.g. page 3 of 18, line 53-54). This is quite untrue -- even if Eq. 2 of their paper is considered, where A is a "stochastic matrix" (as is the case in this paper), this equation perfectly describes the dynamics of a discrete-time, discrete-space Markov chain, which is not deterministic. The fact that, given an initial condition, the evolution described by Eq. 2 is completely deterministic is to be viewed in an ensemble-averaged way. However, given an initial condition, two independent "realizations" of this Markov chain will evolve differently and stochastically, and can eventually end up in two different states of consensus. I strongly recommend that the authors suitably incorporate these changes in their discussion of earlier work, and motivation for their own work. I further note that this point does not take away the validity of their "noisy" model, because the noise plays the role of an external driving force that never allows the system to enter an "absorbing" consensus state and instead, allows for a distribution of opinions, which is what one would assume in a realistic model of opinion dynamics.

2. When the adjacency/trust matrix is undirected and any two individuals have a symmetric influence on one another, the measure of opinion diversity provided by the authors seems to be fine, since one would expect only real eigenvalues. However, in the general (and more commonly expected) case, A would not be symmetric and there doesn't seem to be an obvious reason to believe that all its eigenvalues will remain real. In this light, how does the discussion about a (real) upper bound to the opinion diversity remain valid? If a small argument can be made for why the eigenvalues might remain real, or why the upper bound is a real quantity, it should be added to the work. It also seems to me like the adjacency/trust matrix A can allow for negative values, which would act as a negative influence of one person's opinion on the other. This could also be a physically relevant situation. Do all the results go through if certain elements of A are negative? Overall, it would be nice to have a more complete discussion about the limits and regimes of validity of the results.

Minor comments:

- (i) Page 3 of 18, line 38: weird citation error. Also, in this line, I suppose there is no reason to call A the trust matrix, and instead just call it the weighted adjacency matrix, given that A is being called adjacency matrix later.
- (ii) Page 3 of 18, line 52: typo while spelling 'homophily'.
- (iii) Page 5 of 18, equation 3: the identity matrix should be written in the $\mathbb{1}$ format.
- (iv) Page 11 of 18, line 48: the spelling of 'experienced' is incorrect.
- (v) Page 13 of 18, line 44: citation error.
- (vi) Page 16 of 18, line 40: missing "we" in "we do not have an analytical expression..."

(vii) All the figures will benefit from a much larger font size for the labels of axes, as well as the tick labels.

Review form: Reviewer 2

Is the manuscript scientifically sound in its present form?

Yes

Are the interpretations and conclusions justified by the results?

Yes

Is the language acceptable?

Yes

Do you have any ethical concerns with this paper?

No

Have you any concerns about statistical analyses in this paper?

No

Recommendation?

Accept as is

Comments to the Author(s)

The authors of the present paper study the impact of noise and topology on opinion diversity in social networks. They extend well known models including noise. They also derive analytical expressions to quantify opinion diversity, finding that the latter decreases as communities are broken down. Predictions are compared with synthetic and online news data.

I think that the paper is interesting, very detailed and well written. It presents original sound results and new quantitative measures that allow a better understanding of opinion dynamics on networks. I believe that this paper could be of great interest to a general audience, therefore I definitely suggest its publication in the present form.

Decision letter (RSOS-201943.R0)

Dear Dr Livan,

On behalf of the Editors, we are pleased to inform you that your Manuscript RSOS-201943 "The Impact Of Noise And Topology On Opinion Dynamics In Social Networks" has been accepted for publication in Royal Society Open Science subject to minor revision in accordance with the referees' reports. Please find the referees' comments along with any feedback from the Editors below my signature.

Please submit your revised manuscript and required files (see below) no later than 7 days from today's (ie 16-Feb-2021) date. Note: the ScholarOne system will 'lock' if submission of the revision is attempted 7 or more days after the deadline. If you do not think you will be able to meet this deadline please contact the editorial office immediately.

on behalf of Professor Marta Kwiatkowska (Subject Editor)
openscience@royalsociety.org

Reviewer comments to Author:

Reviewer: 1
Comments to the Author(s)

The authors study models of opinion dynamics on social networks and aim to quantify the effects of noise on the outcome, as well as the impact of network topology. They focus on two rather well studied models -- 1) DeGroot, and 2) Friedkin-Johnsen, and introduce their noisy counterparts, which they study on Erdos-Renyi random networks. They define a useful quantity termed 'opinion diversity', for which they provide a general formula for arbitrary undirected graphs, and show that it is a quantity of interest by working with a real dataset, and improving upon the previous modelling attempt on this dataset, by incorporating opinion diversity in the model. Overall, I think the authors have considered an interesting problem, and have presented a nice analysis -- with useful analytical results, followed by numerical verification and an application to real data. Keeping in mind the aim and scope of the Royal Society Open Science journal, I am happy to recommend this paper titled "The Impact Of Noise And Topology On Opinion Dynamics In Social Networks" for publication provided that the authors address the comments, and fix some minor typos/citation errors as mentioned below:

1. Throughout the work, the authors point to the fact that so far, most models of opinion dynamics that have been studied have been deterministic in nature (e.g. page 3 of 18, line 53-54). This is quite untrue -- even if Eq. 2 of their paper is considered, where A is a "stochastic matrix" (as is the case in this paper), this equation perfectly describes the dynamics of a discrete-time, discrete-space Markov chain, which is not deterministic. The fact that, given an initial condition,

the evolution described by Eq. 2 is completely deterministic is to be viewed in an ensemble-averaged way. However, given an initial condition, two independent "realizations" of this Markov chain will evolve differently and stochastically, and can eventually end up in two different states of consensus. I strongly recommend that the authors suitably incorporate these changes in their discussion of earlier work, and motivation for their own work. I further note that this point does not take away the validity of their "noisy" model, because the noise plays the role of an external driving force that never allows the system to enter an "absorbing" consensus state and instead, allows for a distribution of opinions, which is what one would assume in a realistic model of opinion dynamics.

2. When the adjacency/trust matrix is undirected and any two individuals have a symmetric influence on one another, the measure of opinion diversity provided by the authors seems to be fine, since one would expect only real eigenvalues. However, in the general (and more commonly expected) case, A would not be symmetric and there doesn't seem to be an obvious reason to believe that all its eigenvalues will remain real. In this light, how does the discussion about a (real) upper bound to the opinion diversity remain valid? If a small argument can be made for why the eigenvalues might remain real, or why the upper bound is a real quantity, it should be added to the work. It also seems to me like the adjacency/trust matrix A can allow for negative values, which would act as a negative influence of one person's opinion on the other. This could also be a physically relevant situation. Do all the results go through if certain elements of A are negative? Overall, it would be nice to have a more complete discussion about the limits and regimes of validity of the results.

Minor comments:

- (i) Page 3 of 18, line 38: weird citation error. Also, in this line, I suppose there is no reason to call A the trust matrix, and instead just call it the weighted adjacency matrix, given that A is being called adjacency matrix later.
- (ii) Page 3 of 18, line 52: typo while spelling 'homophily'.
- (iii) Page 5 of 18, equation 3: the identity matrix should be written in the $\mathbb{1}$ format.
- (iv) Page 11 of 18, line 48: the spelling of 'experienced' is incorrect.
- (v) Page 13 of 18, line 44: citation error.
- (vi) Page 16 of 18, line 40: missing "we" in "we do not have an analytical expression..."
- (vii) All the figures will benefit from a much larger font size for the labels of axes, as well as the tick labels.

Reviewer: 2

Comments to the Author(s)

The authors of the present paper study the impact of noise and topology on opinion diversity in social networks. They extend well known models including noise. They also derive analytical expressions to quantify opinion diversity, finding that the latter decreases as communities are broken down. Predictions are compared with synthetic and online news data.

I think that the paper is interesting, very detailed and well written. It presents original sound results and new quantitative measures that allow a better understanding of opinion dynamics on networks. I believe that this paper could be of great interest to a general audience, therefore I definitely suggest its publication in the present form.

===PREPARING YOUR MANUSCRIPT===

===PREPARING YOUR REVISION IN SCHOLARONE===

- Any electronic supplementary material (ESM).
- If you are requesting a discretionary waiver for the article processing charge, the waiver form must be included at this step.
- If you are providing image files for potential cover images, please upload these at this step, and inform the editorial office you have done so. You must hold the copyright to any image provided.
- A copy of your point-by-point response to referees and Editors. This will expedite the preparation of your proof.

- Ensure that your data access statement meets the requirements at <https://royalsociety.org/journals/authors/author-guidelines/#data>. You should ensure that you cite the dataset in your reference list. If you have deposited data etc in the Dryad repository, please only include the 'For publication' link at this stage. You should remove the 'For review' link.
- If you are requesting an article processing charge waiver, you must select the relevant waiver option (if requesting a discretionary waiver, the form should have been uploaded at Step 3 'File upload' above).
- If you have uploaded ESM files, please ensure you follow the guidance at <https://royalsociety.org/journals/authors/author-guidelines/#supplementary-material> to include a suitable title and informative caption. An example of appropriate titling and captioning may be found at https://figshare.com/articles/Table_S2_from_Is_there_a_trade-off_between_peak_performance_and_performance_breadth_across_temperatures_for_aerobic_scope_in_teleost_fishes_/3843624.

Author's Response to Decision Letter for (RSOS-201943.R0)

See Appendix A.

RSOS-201943.R1 (Revision)

Review form: Reviewer 1

Is the manuscript scientifically sound in its present form?

Yes

Are the interpretations and conclusions justified by the results?

Yes

Is the language acceptable?

Yes

Do you have any ethical concerns with this paper?

No

Have you any concerns about statistical analyses in this paper?

No

Recommendation?

Accept as is

Comments to the Author(s)

The authors have considered all my suggestions and have provided a satisfactory response to all. I recommend publication of this paper in its current form.

Review form: Reviewer 2

Is the manuscript scientifically sound in its present form?

Yes

Are the interpretations and conclusions justified by the results?

Yes

Is the language acceptable?

Yes

Do you have any ethical concerns with this paper?

No

Have you any concerns about statistical analyses in this paper?

No

Recommendation?

Accept as is

Comments to the Author(s)

The paper has been improved according to the reviewers' comments, I suggest to accept the paper for publication

Decision letter (RSOS-201943.R1)

Dear Dr Livan,

It is a pleasure to accept your manuscript entitled "The Impact Of Noise And Topology On Opinion Dynamics In Social Networks" in its current form for publication in Royal Society Open

Science. The comments of the reviewer(s) who reviewed your manuscript are included at the foot of this letter.

You can expect to receive a proof of your article in the near future. Please contact the editorial office (openscience@royalsociety.org) and the production office (openscience_proofs@royalsociety.org) to let us know if you are likely to be away from e-mail contact – if you are going to be away, please nominate a co-author (if available) to manage the proofing process, and ensure they are copied into your email to the journal.

on behalf of Prof Marta Kwiatkowska (Subject Editor)
openscience@royalsociety.org

Reviewer comments to Author:
Reviewer: 1

Comments to the Author(s)
The authors have considered all my suggestions and have provided a satisfactory response to all. I recommend publication of this paper in its current form.

Reviewer: 2

Comments to the Author(s)
The paper has been improved according to the reviewers' comments, I suggest to accept the paper for publication

Appendix A

Response to the reviews of submission RSOS-201943 “The Impact Of Noise And Topology On Opinion Dynamics In Social Networks”

We thank the Reviewers for their comments and feedback, which gave us the opportunity to improve our paper. In the following, we provide point-by-point replies to all comments and issues raised by Reviewer #1 (Reviewer #2 did not raise any issues that required changes to be made to the paper).

Reviewer #1

Throughout the work, the authors point to the fact that so far, most models of opinion dynamics that have been studied have been deterministic in nature (e.g. page 3 of 18, line 53-54). This is quite untrue – even if Eq. 2 of their paper is considered, where A is a “stochastic matrix” (as is the case in this paper), this equation perfectly describes the dynamics of a discrete-time, discrete-space Markov chain, which is not deterministic. The fact that, given an initial condition, the evolution described by Eq. 2 is completely deterministic is to be viewed in an ensemble-averaged way. However, given an initial condition, two independent “realizations” of this Markov chain will evolve differently and stochastically, and can eventually end up in two different states of consensus. I strongly recommend that the authors suitably incorporate these changes in their discussion of earlier work, and motivation for their own work. I further note that this point does not take away the validity of their “noisy” model, because the noise plays the role of an external driving force that never allows the system to enter an “absorbing” consensus state and instead, allows for a distribution of opinions, which is what one would assume in a realistic model of opinion dynamics.

It is true that opinion dynamics have been studied in non-deterministic contexts, particularly when using mean-field approaches. That being said, to the best of our knowledge agent-based models have not generally included exogenous noise to represent free will, with notable exceptions that are discussed in the paper. While it is the case that the formula in Eq.2 describes a discrete time Markov Chain when the elements of A are transition probabilities, in our scenario, A does not represent transition probabilities but rather weights of a weighted-averaging process. Given an initial condition, the system is guaranteed to evolve in the same way each time. We have added a sentence to clarify this point in Section 3.

When the adjacency/trust matrix is undirected and any two individuals have a symmetric influence on one another, the measure of opinion diversity provided by the authors seems to be fine, since one would expect only real eigenvalues. However, in the general (and more commonly expected) case, A would

not be symmetric and there doesn't seem to be an obvious reason to believe that all its eigenvalues will remain real. In this light, how does the discussion about a (real) upper bound to the opinion diversity remain valid? If a small argument can be made for why the eigenvalues might remain real, or why the upper bound is a real quantity, it should be added to the work. It also seems to me like the adjacency/trust matrix A can allow for negative values, which would act as a negative influence of one person's opinion on the other. This could also be a physically relevant situation. Do all the results go through if certain elements of A are negative? Overall, it would be nice to have a more complete discussion about the limits and regimes of validity of the results.

We thank the Reviewer for their comment. While we are unable to provide a definitive proof about the realness of the opinion diversity in the case of asymmetric A as given by Eq. (28), we provide a discussion in appendix A on why it appears that d is indeed real valued irrespective of whether A is symmetric or asymmetric. Indeed, the upper bound we derive for d (Eq. (34)) is real-valued, which suggests that the full expression of Eq. (28) is real-valued as well.

Moreover, we make no assumptions about the sign of the adjacency matrix entries, other than requiring it to be sub-stochastic. Therefore the presence of negative influence between nodes would change neither the realness of the eigenvalues (in the symmetric case) nor the expression we derive for d .

Minor comments:

- (i) Page 3 of 18, line 38: weird citation error. Also, in this line, I suppose there is no reason to call A the trust matrix, and instead just call it the weighted adjacency matrix, given that A is being called adjacency matrix later.*
- (ii) Page 3 of 18, line 52: typo while spelling 'homophily'.*
- (iii) Page 5 of 18, equation 3: the identity matrix should be written in the $\mathbb{1}$ format.*
- (iv) Page 11 of 18, line 48: the spelling of 'experienced' is incorrect.*
- (v) Page 13 of 18, line 44: citation error.*
- (vi) Page 16 of 18, line 40: missing "we" in "we do not have an analytical expression..."*
- (vii) All the figures will benefit from a much larger font size for the labels of axes, as well as the tick labels.*

We thank the reviewers for pointing out the typos and other minor errors. We fixed these issues.